# Symptomatic Intraosseous Lipoma of the Calcaneum

**DOI:** 10.3390/diagnostics11122243

**Published:** 2021-11-30

**Authors:** Adyb Adrian Khal, Razvan Catalin Mihu, Calin Schiau, Bogdan Fetica, Gheorghe Tomoaia, Manuel Vergillos Luna

**Affiliations:** 1Department of Orthopaedics and Traumatology, Iuliu Hatieganu University of Medicine and Pharmacy, 400000 Cluj-Napoca, Romania; razvan_mihu@yahoo.com (R.C.M.); tomoaia2000@yahoo.com (G.T.); 2Regina Maria Private Health Care Network, 400117 Cluj-Napoca, Romania; calin.schiau@yahoo.com; 3Department of Orthopaedics, Lenval University Children’s Hospital, 06200 Nice, France; 4Department of Radiology, Iuliu Hatieganu University of Medicine and Pharmacy, 400000 Cluj-Napoca, Romania; 5Department of Pathology, Oncology Institute Ion Chiricuţă, 400015 Cluj-Napoca, Romania; feticab@yahoo.com; 6Department of Orthopaedics, Regina Margherita Paediatric Hospital, 10126 Torino, Italy; m.vergillosluna@gmail.com

**Keywords:** intraosseous lipoma, pain, curettage, bone cement

## Abstract

Intraosseous lipomas are rare bone lesions that can affect any part of the skeleton. In the calcaneum, they are, generally, asymptomatic, but in some cases, patients may complain of pain, swelling or tenderness. Well-conducted radiography and MRI examinations can lead to an accurate diagnosis. In most cases, patients could benefit from conservative means of treatment, but in long-lasting symptomatic cases, surgical treatment may be a good option. The purpose of this article is to increase clinicians’ awareness of this lesion as a possible cause of heel pain and to describe a case of a symptomatic intraosseous lipoma of the calcaneum who underwent curettage and bone cement filling after failure of conservative treatment.

## 1. Introduction

The first indexed description of an intraosseous lipoma dates back to 1976 [1] but cases have been reported as early as 1880 [2,3]. At present, there are no clear epidemiological data about benign bone tumors [4,5], but the total incidence of intraosseous lipomas is estimated to be between 0.02 and 0.1% of the total incidence of primary benign bone tumours [4,6].

The most common location is the lower limb, usually within the calcaneum, femur, tibia, and fibula, but intraosseous lipomas may occur in any bone of the body [4,7,8,9].

Most cases are asymptomatic and are usually incidentally discovered on radiographs performed for unrelated disorders [7]. However, patients may sometimes complain of heel pain that appears after physical effort or minor trauma [7,10,11].

These lesions are frequently misdiagnosed [11]. Clinically, plantar fasciitis [2], Haglund disease [12], or stress fractures [2] have similar symptoms, while radiologically, simple bone cysts [7], pseudo benign tumours [13], or osteoblastomas [2] have similar appearances. However, if a plain radiography and an MRI exam are thoroughly conducted, a precise diagnosis is ascertained [3,14].

The decision whether to treat these lesions surgically or the preference towards a conservative treatment is controversial [5,13,15]. The effectiveness of intralesional resection followed by autograft [16], artificial bone substitute [7,11,13,17], or cement filling [18] is widely accepted.

In this report, we describe a case of a symptomatic, painful, intraosseous lipoma of the calcaneum who underwent curettage and cement filling after failure of several months of conservative treatment.

## 2. Case Presentation

A 40-year-old male engineer, former professional rugby player, was referred to our clinic with a left heel inflammatory pain that was worsening during jogging or trailing. The symptoms started seven months before and the patient presented to a regional local hospital for investigations where a plain radiography was performed and a simple bone cyst diagnosis was suspected. Patient was recommended a break from physical activity for six months and non-steroid anti-inflammatories drugs (NSAIDS) to ameliorate pain. The pain increased gradually and he started to complain of swelling. The pain was hardly controlled with NSAIDS and non-morphinic analgesics.

Our clinical examination revealed a mild tenderness in the posterior foot, including the ankle and the heel, without evidence of a palpable mass. Ankle and subtalar joint mobilities were limited. The laboratory blood tests and urine analysis results were normal.

Radiography revealed a benign-appearing bone lesion of 16 × 19 mm within the anteroinferior part of the calcaneum which was well defined, radiolucent, almost entirely homogeneous with a small central sclerotic focus-“Cockade sign” [2,14], describing the classical appearance of a calcaneal intraosseous lipoma (Figure 1). We also performed and MRI exam that showed a focal lesion, hyperintense on both T1 and T2 weighted images, and isointense with fatty tissues (Figure 2a,b). There was a discreet focal attenuation in the center of the lesion, on T2* sequence, suggestive for focal calcification (Figure 2c). The MRI aspect corresponded to a Milgram type II intraosseous lipoma (predominantly fatty lesions with central necrosis/calcifications/ ossifications) [3].

Surgery was performed and a direct lateral approach to the calcaneum was chosen. The saphenous nerve and the long peroneus tendon were reclined superiorly and distally (Figure 3a). A bone window was performed immediate distally to the lateral tubercle of the calcaneum. Aggressive curettage of the lesion was carried out and the intralesional samples were send to the histopathological exam (Figure 3b). The cavity was filled with a medium viscosity poly (methyl methacrylate)-based bone cement with gentamycin. No perioperative complications occurred.

The histopathological exam (Figure 4) and immunohistochemistry (S100 and Vimentin positive) confirmed the diagnosis, a stage II Milgram intraosseous lipoma.

Postoperatively, the pain subsided completely. The patient was discharged the second day after the surgery. Total weight bearing was allowed without crutches. No cast or brace was applied. Rehabilitation was conducted for two weeks in order to encourage full ankle mobilities.

In the follow-up, the patient was evaluated every six weeks during the first three months and every six months until the first two years after surgery. Starting from the first postoperative consultation at six weeks, the ankle joint mobilities were normal, the patient was painless, and radiological examination showed no modifications. At the time of the last follow-up at 24 months, no modification and no recurrence were observed (Figure 5a,b) and the patient returned to work six weeks after the surgery.

## 3. Discussion

Intraosseous lipomas are very rare lesions [4,6]. Approximately 300 cases have been reported in the literature until 2014 [2] but, due to the development of diagnostic techniques, the number of reported cases have increased.

The average age at the time of diagnosis ranges from 30 to 60 years [6,8] and most cases are asymptomatic, although pain, swelling and tenderness can be present in symptomatic patients at the time of diagnosis [11,12,13,17]. In our case, symptoms started with pain that lasted almost 1 year until a mild tenderness and limited mobilities occurred.

The radiological appearance of intraosseous lipomas is often uncharacteristic and can be confused easily with other bone lesions such as chondroid tumors, aneurysmal bone cysts, fibrous dysplasia, bone infarcts and liposclerosing myxofibrous tumours [19]. Moreover, benign radiological findings do not raise the curiosity of the surgeon for a more extensive imagistical exam. Usually, the hypothesis of an intraosseous lipoma of the calcaneum is based on a CT scan or MRI examination. Radiological features often change with the histological stage, ranging from radiolucent lesions with a thin sclerotic border to radiodense lesions with a thick sclerotic border [3]. Milgram classified bone lipomas are based on the degree of involution [3]: stage I lesions are composed of mature fat cells that resemble those of the subcutaneous tissue; stage II lesions demonstrate mainly mature fat cells associated with some necrotic foci, foamy macrophages and calcification; and stage III lesions show necrotic fat with focal calcifications and cystic degeneration. The Cockade sign, seen as a well-defined lytic lesion with a central calcification, is the classic appearance of calcaneal intraosseous lipomas and may be present in Milgram II lesions [7,14]. Common location of bone lipomas include the femur, tibia, and humerus but the most frequent is the calcaneum (32% overall) [4,7,8].

Usually, diagnosis is easily-defined when MRI and CT scans are performed [8]. If radiological features are not conclusive, a biopsy is needed to provide useful information for a correct diagnosis [7,8]. Morphopathologically, intraosseous lipoma can be a challenging diagnosis due to the adipose tissue that is a part of the marrowfat [6]. We think that “fat in lesion” versus “normal fat” need a thoroughly conducted correlation between clinical, radiological, and histopathological data. However, the key of the diagnosis are the imaging features and confidence that the surgeon biopsied the lesion. In more difficult cases when radiological and histological data are not consistent, or in order to exclude an intraosseous liposarcoma, the MDM2 analysis may be a valuable tool to confirm the diagnosis [20]. No other particular histological features are required except the replacement of bone tissue with adipose tissue.

Conservative treatment consisting of clinical and radiological follow-up is advocated in asymptomatic patients without impending pathologic fracture or suspicion of malignancy [8,15]. In our case, conservative means of treatment were not sufficient enough to ameliorate the symptoms. The same problems emerged regarding the conservative treatment described Ulucay [5] in his 22 calcaneal lipoma case series which were resistant to conservative treatment. Lesions mainly consisted of stage I (11/22) and stage 2 lesions (9/22), and all patients went on to achieve pain improvement at one-year follow-up after curettage of the lesion and filling with homologous bone. No postoperative complications were reported. Asymptomatic patients could benefit from conservative means of treatment and a watchful waiting approach, while patients that accuse chronic or recurrent heel pain can undergo surgery to prevent impairment of daily activities or complications such as pathologic fracture of the calcaneum [13].

There is no consensus regarding the surgical treatment of symptomatic bone lipomas, nor is there consensus concerning when a surgical procedure should be performed. In case of failure of conservative treatment, common surgical options include curettage with or without filling of the cavity with different substitutes [7,11,21]. Reconstruction of the cavity with bone cement [18], autologous bone grafts [5] or bone substitutes have been described by other authors [7,11,21], all having reported good results. Narang [11] and Muramatsu [13] utilized bone substitutes to fill a total of seven intraosseous calcaneal lipomas. Both studies showed good filling of the cavity with resolution of pain and no postoperative complications. Kang [7] reported a total of 20 bone lipomas treated with curettage and filling of the cavity with allograft bone chips, of which 11 were symptomatic. Pain was resolved in 7 out of 11 cases and no complications arose during follow-up. Greenspan [21] also presented in 1997 six cases of intraosseous lipomas of the calcaneus, three of which underwent curettage and filling with bone chips with resolution of the pain. Mawardi [18] reported a case of heel pain associated with a lytic lesion of the calcaneus. Pathologic report confirmed the diagnosis of intraosseous lipoma; the patient remained pain-free and deambulation was regained one month after bone curettage and filling of the cavity with bone cement. Whatever material used, the filling will provide mechanical and structural support [22]. In our case, we preferred bone cement in order to allow immediate total weight bearing, since it is also easy-reproductible and was available at low costs. Moreover, its cytotoxic and thermal effects to tumor cell and early recurrence detection in the follow-up makes it a good option [22]. When the risk of fracture was higher [16,17], sometimes associated internal fixation of the calcaneus was used. Two case reports [16,17] are available in the literature where curettage and bone grafting were associated with internal fixation due to increased risk of fracture. Both cases went on to heal without complications and no signs of tumor recurrence were present at final follow-up.

This study is a case report and it was subject to inherent limitations and biases. The study was retrospective, the technique of reconstruction was not randomized, and the preference of the surgeon may have contributed to a selection bias. However, bone lipomas are rare lesions, and we report a case with a good follow-up when the literature convinces us about the current controversy regarding the surgical treatment and time when such treatment should be performed.

## 4. Conclusions

Despite the low prevalence of lipomas of the calcaneum, physicians need to be aware about this lesion that can cause heel pain. Intraosseous lipomas are more often asymptomatic, but patients can present chronic heel pain, local inflammation signs, and even edema. Being able to diagnose an intraosseous lipoma through MRI or CT scan imaging and offering the best treatment options is essential for the health of future patients. It is our opinion that bone cement is cheap, easy to handle, and common enough to be easily available at most hospitals, making it a good option to fill lytic lesions after the curettage of bone lipomas while consenting immediate full-weight bearing.

## Figures and Tables

**Figure 1 diagnostics-11-02243-f001:**
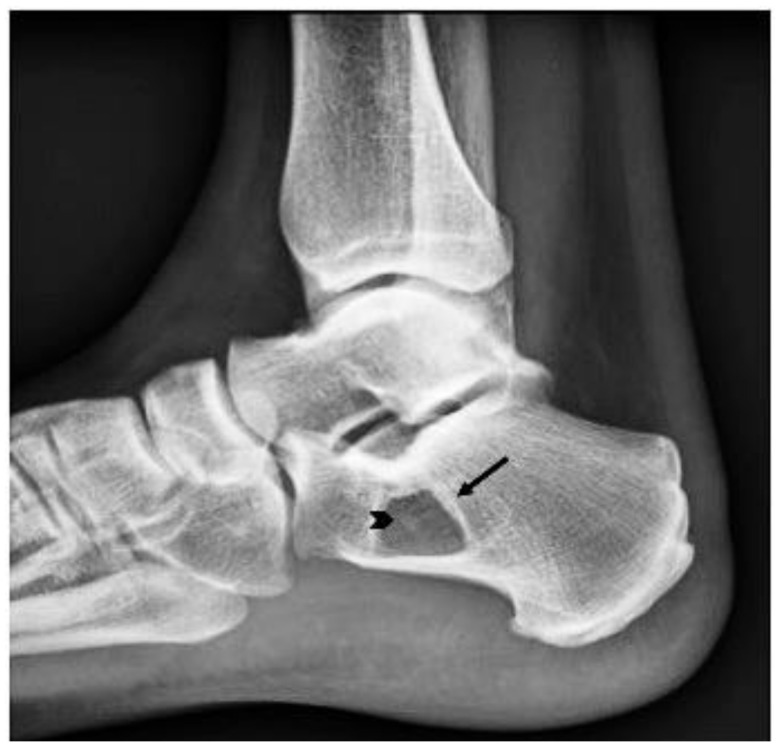
Plain radiography, lateral projection. Lytic calcaneus lesion (arrow). Small central sclerotic focus (arrowhead)-“Cockade sign” [14].

**Figure 2 diagnostics-11-02243-f002:**
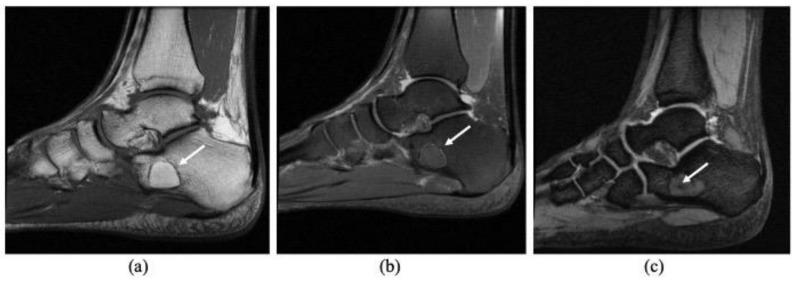
MRI sagittal views. T1 FSE (**a**), PD FSE FS (**b**) and T2* (**c**) sequences. MRI reveals an intraosseous mass (arrows), with homogeneous fat suppression (**a**,**b**). Discreet focal attenuation within the center of the lesion suggestive for focal calcification (**c**).

**Figure 3 diagnostics-11-02243-f003:**
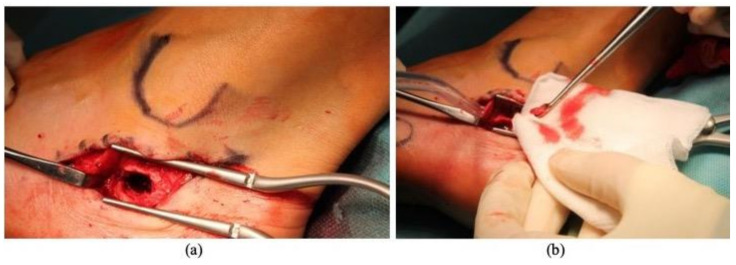
Intraoperatory images during a curettage (**b**) of an intraosseous lipoma of the calcaneum through a lateral approach. The peroneus tendons were reclined and a bone window, immediately distal to the lateral processus of the calcaneum (**a**), was performed in order to get access to the lesion.

**Figure 4 diagnostics-11-02243-f004:**
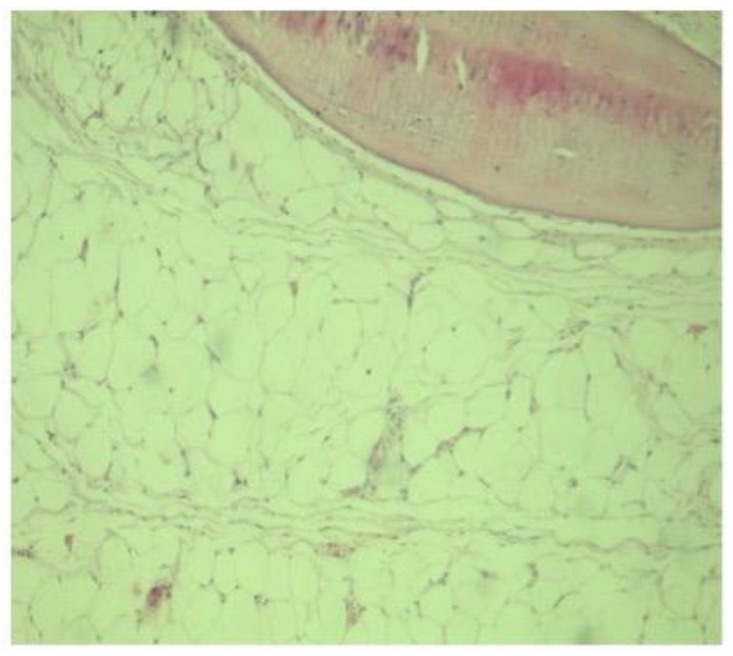
Histological specimen showing adipose tissue (lobulated with areas of mature lipocytes) in apposition with small areas of medullary trabecular bone (upper right corner).

**Figure 5 diagnostics-11-02243-f005:**
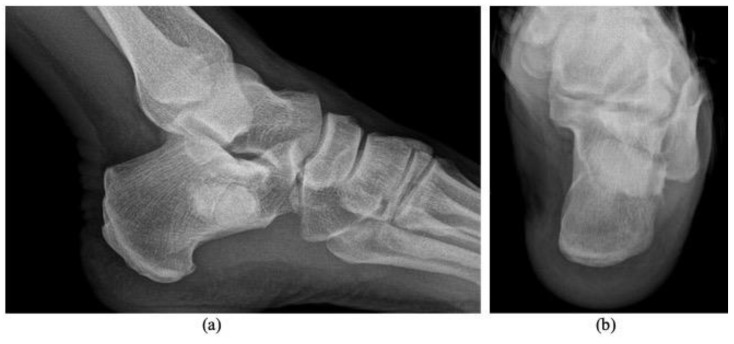
Calcaneum AP (**a**) and axial (**b**) Radiographs show a two years follow-up result of an intraosseous lipoma that underwent curettage and cement filling. No sign of local recurrence is present.

## Data Availability

On request from the corresponding author, the data are not publicly available due to privacy and ethical reasons.

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
