# Peer review of "Symptomatic Intraosseous Lipoma of the Calcaneum"

_diagnostics, 2021, doi:10.3390/diagnostics11122243_

Round 1
Reviewer 1 Report
The article describes the case; this description is valuable and should be published in "Diagnostics".
However, I believe that the article requires a thorough revision, mainly due to the incorrect structure of the text.
I have no objections to the Material and Results chapter. He believes that all fragments related to the shoulder must be removed. It has nothing to do with the title of the article or with the presented lipoma problem.
In the x-ray image, I propose to increase the contrast in order to better visualize the sclerotic features of the lipoma rim and "bows". This is a very important differentiation feature.
My biggest comments are on the Discussion chapter, which is completely incorrect. This chapter is completely subject to change. Already from line 111, with few exceptions, the text should be placed in the Introduction chapter, i.e. showing the problem already discussed in the literature. The authors show not a meta-analysis of lipoma occurrence, but present a single surgical case. The few lines that are the correct form of discussing one's own research with world literature may remain in the current discussion. The authors should therefore conduct a proper discussion. On lines 29 and 115 there are repetitions of the text.
The text is linguistically understandable and pleasant to read. On line 187, instead of "this lesion as a cause of heel pain" please write "can cause heel pain".
In Conclusion, I don't know why the authors have made the aspect of cement so emphasized. After all, there is no analysis of other fillings in the article.
Fig. 4. I cannot see the features of spongy bone tissue, possibly undifferentiated bone tissue. Proposes to trim to adipose tissue.
Author Response
Article Title: Symptomatic intraosseous lipoma of the calcaneum
Manuscript ID: diagnostics-1469627
Reviewer #1:
Reviewer(s)' Comments to Author:
- The article describes the case; this description is valuable and should be published in "Diagnostics". However, I believe that the article requires a thorough revision, mainly due to the incorrect structure of the text.
- I have no objections to the Material and Results chapter. He believes that all fragments related to the shoulder must be removed. It has nothing to do with the title of the article or with the presented lipoma problem.
- In the x-ray image, I propose to increase the contrast in order to better visualize the sclerotic features of the lipoma rim and "bows". This is a very important differentiation feature.
- My biggest comments are on the Discussion chapter, which is completely incorrect. This chapter is completely subject to change. Already from line 111, with few exceptions, the text should be placed in the Introduction chapter, i.e. showing the problem already discussed in the literature. The authors show not a meta-analysis of lipoma occurrence, but present a single surgical case. The few lines that are the correct form of discussing one's own research with world literature may remain in the current discussion. The authors should therefore conduct a proper discussion. On lines 29 and 115 there are repetitions of the text.
- The text is linguistically understandable and pleasant to read. On line 187, instead of "this lesion as a cause of heel pain" please write "can cause heel pain".
- In Conclusion, I don't know why the authors have made the aspect of cement so emphasized. After all, there is no analysis of other fillings in the article.
- 4. I cannot see the features of spongy bone tissue, possibly undifferentiated bone tissue. Proposes to trim to adipose tissue.
Dear Reviewer,
Thank you for giving us the opportunity to submit the revised draft of the Manuscript diagnostics-1469627, entitled „ Symptomatic intraosseous lipoma of the calcaneum” for publication in the journal of Diagnostics. We appreciate the time and effort that you dedicated to providing feedback on our manuscript and are grateful for the insightful comments on and valuable improvements to our paper. We have incorporated the suggestions made by the reviewer. Those changes are marked up using the “Track Changes” function (red color). Please see below, for a point-by point response to the comments and concerns.
- Thank you for your appreciation and comments. The manuscript was modified according to your suggestions.
- Thank you for this comment. All information concerning the shoulder injuries were removed from the manuscript.
- The contrast and the sharpness of the X-ray image was changed in order to better visualize the sclerotic rim of the intraosseous lipoma.
- Thank you for this precious comment. The first paragraph from the initial Discussions chapter was moved to the Introduction. The actual Discussions chapter was restructured according your valuable comments and the references was reordered. We tried to better compare our case report with the experience of other authors who have already published their means of diagnostic and treatment regarding the bone lipomas and to conduct a proper discussion. We checked the manuscript for repetition of phrases and we removed it from the text.
- Thank you very much for your appreciation. We modified the phrase accordingly.
Line 278 – 279: “Despite the low prevalence of lipomas of the calcaneum, physicians need to be aware about this lesion that can cause heel pain. »
- A short analysis of different means of bone filling was added in the manuscript. We think that is important to notice the readers that bone cement is a sustainable mean of reconstruction of the bone defect after curettage, permitting patients to full weight bearing immediately after the surgery and available at low costs. In term of follow-up, similar results described other authors, but full weight bearing was not allowed immediately postoperative.
- Thank you for this comment. The diagnosis was made by pathologists at our Institution. Unfortunately, we cannot provide better morpho pathological images in this short time permitted for the revision. However, we tried to draw attention of adipose tissue in apposition with a bone trabeculae in the already provided figure. We believe that including more images of histhopathology samples would not improve the quality of the paper. As the aim of this case report is to describe the symptoms, radiological diagnostic and surgical treatment after failure of the conservative treatment, we think that x-rays and MRIs of the involved limb would be more helpful.
Reviewer 2 Report
The authors Khal and colleagues report an un common presentation of intraosseous lipoma affecting the calcaneum. In particular the authors described the case presentation of this unusual lesion including X-ray, MRI analysis, immunohistochemical and surgery.
The paper is interesting and focus on a rare entity
The following should be addressed:
- Immunohistochemical images of S100 and Vimentin should be included.
- In order to exclude the diagnosis of a intraosseus liposarcoma MDM2 expression analysis should be included. If is not possible the authors should referred in the manuscript the following work: “Current classification, treatment options, and new perspectives in the management of adipocytic sarcomas”. Onco Targets Ther. 2016 Oct 11;9:6233-6246. doi: 10.2147/OTT.S112580. PMID: 27785071; PMCID: PMC5067014.
- The following reference should be reported “A rare case of intraosseous lipoma involving the sphenoclival region”. Neuroradiol J. 2012 Dec 20;25(6):680-3. doi: 10.1177/197140091202500607. Epub 2012 Dec 20. PMID: 24029181.
- Study limitations should be described
Author Response
Article Title: Symptomatic intraosseous lipoma of the calcaneum
Manuscript ID: diagnostics-1469627
Reviewer #2:
Reviewer(s)' Comments to Author:
The authors Khal and colleagues report an un common presentation of intraosseous lipoma affecting the calcaneum. In particular the authors described the case presentation of this unusual lesion including X-ray, MRI analysis, immunohistochemical and surgery. The paper is interesting and focus on a rare entity
- The following should be addressed: Immunohistochemical images of S100 and Vimentin should be included.
- In order to exclude the diagnosis of a intraosseus liposarcoma MDM2 expression analysis should be included. If is not possible the authors should referred in the manuscript the following work: “Current classification, treatment options, and new perspectives in the management of adipocytic sarcomas”. Onco Targets Ther. 2016 Oct 11;9:6233-6246. doi: 10.2147/OTT.S112580. PMID: 27785071; PMCID: PMC5067014.
- The following reference should be reported “A rare case of intraosseous lipoma involving the sphenoclival region”. Neuroradiol J. 2012 Dec 20;25(6):680-3. doi: 10.1177/197140091202500607. Epub 2012 Dec 20. PMID: 24029181.
- Study limitations should be described
Dear Reviewer,
Thank you for giving us the opportunity to submit the revised draft of the Manuscript diagnostics-1469627, entitled „ Symptomatic intraosseous lipoma of the calcaneum” for publication in the journal of Diagnostics. We appreciate the time and effort that you dedicated to providing feedback on our manuscript and are grateful for the insightful comments on and valuable improvements to our paper. We have incorporated the suggestions made by the reviewer. Those changes are marked up using the “Track Changes” function (red color). Please see below, for a point-by point response to the comments and concerns.
- Thank you for this comment. The diagnosis was made by pathologists at our Institution. Unfortunately, we cannot provide Immunohistochemical images of S100 and Vimentin as these samples are not more available. However, we tried to better describe the already provided morphopathological figure explaining the adipose tissue in apposition with bone trabeculae. If necessary, we can remove the immunohistochemistry phrase from the manuscript, although, we believe that including more images of histhopathology samples would not improve the quality of the paper. As the aim of this case report is to describe the symptoms, radiological diagnostic and surgical treatment after failure of the conservative treatment, we think that x-rays and MRIs of the involved limb would be more helpful.
- The suggested references were added to the manuscript.
Line 177 – 179: “In more difficult cases when radiological and histological data are not consistent or in order to exclude an intraosseous liposarcoma, the MDM2 analysis may be a valuable tool to confirm the diagnosis [citation].”
- The suggested references were added to the manuscript.
Line 34 – 35: “The most common location is the lower limb, usually within the calcaneum, femur, tibia and fibula, but intraosseous lipomas may occur in any bone of the body [citations]. “
- Limitations are now included in the discussion section.
Line 270 – 275: “This study is a case report and it was subject to inherent limitations and biases. The study was retrospective, the technique of reconstruction was not randomized and the preference of the surgeon may have contributed to a selection bias. However, bone lipomas are rare lesions, and we report a case with a good follow-up when Literature convinces us about the current controversy regarding the surgical treatment and at what time it should be performed.”
Round 2
Reviewer 1 Report
Interesting case report of a calcaneal lipoma. This study is useful for medics and for researchers of the skeletons of prehistoric populations.